# Spectral Decomposed Variational Inference: A Principled Framework for Posterior Covariance Modeling

## Abstract

The Kullback-Leibler (KL) divergence, the cornerstone of standard Variational Inference (VI), acts as a blunt instrument for shaping posterior geometries, forcing an unfavorable trade-off between expressivity and scalability. This paper challenges this limitation by proposing a paradigm shift: moving variational optimization from the space of distributions into the spectral domain of the posterior covariance. We introduce Spectral Decomposed Variational Inference (SD-VI), a novel framework built upon a new class of objectives. Instead of the monolithic KL penalty, SD-VI enables explicit, fine-grained regularization of the covariance's eigenspectrum. This new objective is optimized by our efficient and provably convergent Proximal Spectral Optimization (PSO) algorithm, which leverages a sequence of analytical spectral shrinkage steps to automatically discover sparse, low-rank posterior structures in a principled manner. We demonstrate the power of SD-VI on two challenging applications, showing that it learns significantly better-calibrated Bayesian Neural Networks and enables more scalable inference in Sparse Gaussian Processes. Our work establishes a new, powerful approach for building more robust and efficient Bayesian models through direct geometric control of uncertainty.

## 1 Introduction

In the era of large-scale deep learning, a critical paradigm shift is underway: from a singular focus on predictive accuracy to a holistic demand for models that are reliable, robust, and trustworthy (Abdar et al., 2020; Gawlikowski et al., 2021). This need has become particularly acute for modern foundation models, where understanding model confidence is non-negotiable for safe deployment in high-stakes domains like medical diagnosis or autonomous systems (Lambert et al., 2022; Bilbrey et al., 2025). Bayesian deep learning provides a principled mathematical framework for this challenge, but it grapples with a fundamental scalability-fidelity trade-off.

The cornerstone of the modern Bayesian approximation, Variational Inference (VI), epitomizes this dilemma (Blei et al., 2017). At one end of the spectrum, Mean-Field VI (MF-VI) is fast and scalable, but its core assumption of parameter independence forces it to systematically underestimate uncertainty, resulting in models that are dangerously overconfident (Giordano et al., 2018; Foong et al., 2019). This is especially problematic as modern neural networks are notoriously poorly calibrated (Guo et al., 2017), a challenge that has spurred research into various mitigation strategies, from post-hoc calibration techniques (Tao et al., 2025) to principled Bayesian approaches like ours. At the other end, methods that capture full-rank correlations are often computationally prohibitive for the billion-parameter models common today.

This has ignited a "Cambrian explosion" of structured VI methods, each proposing a clever mechanism to navigate this treacherous middle ground, such as Normalizing Flows (Rezende & Mohamed, 2015) or low-rank plus diagonal covariances (Dusenberry et al., 2020). However, these methods share a deeper and more subtle flaw: the Evidence Lower Bound (ELBO) objective they optimize inherently rewards complexity and lacks a mechanism to prevent overfitting within the variational family (Turner & Sahani, 2011; Rainforth et al., 2018). The critical question of determining the

*optimal* posterior structure for a given task remains largely an unsolved art, creating a critical gap in developing truly robust models.

In this work, we argue that the monolithic and entangled nature of the KL-divergence is the root of this problem. It penalizes the posterior as a whole, inextricably mixing the penalty on the posterior's volume (governed by eigenvalues) with the penalty on its orientation (governed by eigenvectors). This entanglement acts as a blunt instrument, preventing the fine-grained geometric control needed to learn parsimonious yet expressive structures. To break this impasse, we propose a paradigm shift: instead of optimizing in the space of distributions with a single, entangled metric, we move the optimization directly into the spectral domain of the posterior covariance. This allows us to decompose the problem and gain explicit, separate control over the posterior's eigenvalues and eigenvectors.

This principle is instantiated in our new framework, **Spectral Decomposed Variational Inference (SD-VI)**, which enables principled and fine-grained control over the posterior's geometry. Our contributions are threefold:

1. We derive a novel spectral decomposition of the KL-divergence, revealing how it implicitly and inextricably penalizes the posterior's eigenspectrum. Motivated by this, we propose a flexible class of **Spectrally Decomposed Variational Objective (SD-VO)** that replace the entangled KL penalty with explicit, disentangled regularization of the eigenvalues and eigenvectors.

2. We develop the **Proximal Spectral Optimization (PSO)** algorithm, an efficient and provably convergent method for optimizing our novel objective. Drawing on principles from proximal methods (Parikh & Boyd, 2014), PSO transforms the complex constrained optimization into a sequence of simple, analytical spectral shrinkage operations.

3. Through comprehensive experiments, we demonstrate that SD-VI learns significantly better-calibrated Bayesian Neural Networks and enables more scalable inference in Sparse Gaussian Processes, achieving a superior trade-off between performance and efficiency compared to strong baselines.

## 2 RELATED WORK

Our work is situated at the intersection of structured variational families, alternative variational objectives, and the synthesis of spectral and proximal methods. We position our contributions in relation to these key areas.

**Structured Variational Families.** While most prior work designs more flexible variational *families* to better approximate the posterior, our work fundamentally redesigns the variational *objective* to automatically discover the appropriate structure within a given family. The limitations of the mean-field assumption are well-established (Giordano et al., 2018; Foong et al., 2019). Consequently, a vast body of research has explored more expressive posterior families. These include learnable transformations such as normalizing flows (Rezende & Mohamed, 2015; Papamakarios et al., 2021), efficient geometric constructions such as orthogonal VI, and matrix-variate posteriors with Kronecker-factored covariance (Louizos & Welling, 2016). A particularly popular and pragmatic approach has been the low-rank plus diagonal covariance, a computationally efficient heuristic used in many state-of-the-art Bayesian neural networks (Dusenberry et al., 2020; Wenzel et al., 2020). However, these methods typically still optimize the standard ELBO, forcing a manual and often heuristic choice of structural hyperparameters (e.g., rank, flow depth). As noted by Rainforth et al. (2018), a more expressive family does not guarantee a better posterior. Our SD-VI framework addresses this gap by providing a principled objective that actively prunes non-essential dimensions of posterior uncertainty, thereby grounding the common low-rank heuristic in a formal optimization framework and automating the discovery of an effective rank.

**Alternative Variational Objectives.** Our work contributes to the literature on refining the ELBO objective by proposing a unique alternative to existing strategies. Current approaches typically seek to either refine the ELBO's approximation for non-conjugate models (Komodromos et al., 2024; Polson et al., 2013; Durante & Rigon, 2019), or modify the objective itself. The latter is often achieved by replacing the KL term with other monolithic divergences from the f-divergence family (Wan et al., 2020; Cinquin & Bamler, 2024), or by augmenting the objective with a single additive penalty to control model complexity. While valuable, these methods lack direct and explicit control

over the posterior's geometric structure. In sharp contrast, our SD-VI framework is fundamentally different. Instead of using another monolithic penalty, we replace the entire KL-divergence with a structured, decomposed spectral regularizer, $\Phi(S_q)$. This unique formulation provides disentangled, fine-grained control over both the *amount* of posterior uncertainty (via its eigenvalues) and its *orientation* (via its eigenvectors). Consequently, our approach moves beyond simple complexity control towards a more direct and expressive form of geometric, structural discovery in the posterior.

**Spectral and Proximal Methods in Optimization.** Our PSO algorithm represents a novel synthesis of spectral analysis and proximal optimization, creating a scalable inference engine uniquely tailored for our spectrally-defined objective. Proximal algorithms are a powerful toolkit for non-smooth, non-convex optimization (Parikh & Boyd, 2014), and form the backbone of many state-of-the-art algorithms in sparse learning (Beck & Teboulle, 2009) and low-rank matrix recovery, such as the singular value thresholding algorithm (Cai et al., 2010). While optimization on manifolds offers an alternative for related problems (Wen & Yin, 2013), proximal methods are particularly well-suited for objectives with separable penalties. The use of spectral properties to define priors or analyze models is also common (Bishop, 1999). However, inducing posterior structural sparsity is distinct from methods that induce sparsity in model weights via hierarchical priors (Carvalho et al., 2010) or compression techniques (Louizos et al., 2017). Our goal is not model compression, but the discovery of a parsimonious and well-calibrated posterior uncertainty representation. The primary novelty of PSO is to demonstrate that the proximal operator for our sophisticated spectral regularizer has an elegant, analytical solution via scalar shrinkage on the eigenvalues. This insight is what allows us to bring the full power and robust convergence guarantees of proximal optimization to the heart of structured variational inference, transforming a complex matrix optimization problem into a highly efficient, parallelizable one.

# 3 METHODOLOGY: SPECTRAL DECOMPOSED VARIATIONAL INFERENCE

The Evidence Lower Bound (ELBO) objective in conventional variational inference provides a single, scalar measure of divergence, $D_{\mathrm{KL}}(q||p)$. This monolithic penalty acts as a blunt instrument for shaping the complex geometry of the variational posterior, obscuring fine-grained structural discrepancies within the posterior covariance. Unlike mean-field approaches that neglect correlations, or normalizing flows that offer immense flexibility at the cost of interpretability, we seek a new middle ground: one that provides structured, explicit, and interpretable control over the posterior geometry. To achieve this, we introduce **Spectral Decomposed Variational Inference (SD-VI)**. We shift the arena of variational optimization from the space of distributions to the spectral domain of the covariance matrix—the space of its eigenvalues and eigenvectors. **Figure 1** provides a conceptual overview of our approach. We illustrate how standard VI can get "lost in the foothills" of a complex posterior landscape (a), whereas SD-VI performs a principled "survey" of this landscape by decoupling the control of the posterior's spectral properties (b). This allows it to discover a parsimonious and geometrically-aligned posterior structure that captures the true uncertainty along the "main ridge" of the distribution (c). Our methodology unfolds logically in the following sections, detailing each component of this process.

## 3.1 A SPECTRAL VIEW OF THE KL-DIVERGENCE

Let the variational posterior be $q(\boldsymbol{\beta}) = \mathcal{N}(\boldsymbol{\beta}|\boldsymbol{\mu}, \mathbf{S}_q)$ and the prior be $p(\boldsymbol{\beta}) = \mathcal{N}(\boldsymbol{\beta}|\boldsymbol{\mu}_p, \mathbf{S}_p)$, where our goal is to optimize the variational covariance $\mathbf{S}_q$ to approximate the fixed prior covariance $\mathbf{S}_p$. The KL-divergence between these two multivariate Gaussian distributions is given by:

$$D_{\mathrm{KL}}(q||p) = \frac{1}{2}\left[\log\frac{|\mathbf{S}_p|}{|\mathbf{S}_q|} + \mathrm{Tr}(\mathbf{S}_p^{-1}\mathbf{S}_q) + (\boldsymbol{\mu} - \boldsymbol{\mu}_p)^T\mathbf{S}_p^{-1}(\boldsymbol{\mu} - \boldsymbol{\mu}_p) - p\right] \tag{1}$$

where $p$ is the dimensionality of the parameter vector $\boldsymbol{\beta}$. To understand the geometric implications of this objective, we perform a spectral decomposition of the variational covariance, $\mathbf{S}_q = \mathbf{U}\,\mathrm{diag}(\boldsymbol{\sigma})\mathbf{U}^T$, where $\boldsymbol{\sigma} = (\sigma_1, \ldots, \sigma_p)$ is the vector of eigenvalues (the eigenspectrum) and $\mathbf{U} = [\mathbf{u}_1, \ldots, \mathbf{u}_p]$ is the orthogonal matrix of corresponding eigenvectors. The determinant term $\log|\mathbf{S}_q|$ simplifies to $\sum_{i=1}^{p}\log(\sigma_i)$, revealing a direct logarithmic penalty on the eigenvalues. The trace term, however, couples the eigenvalues and eigenvectors in a more complex manner. By

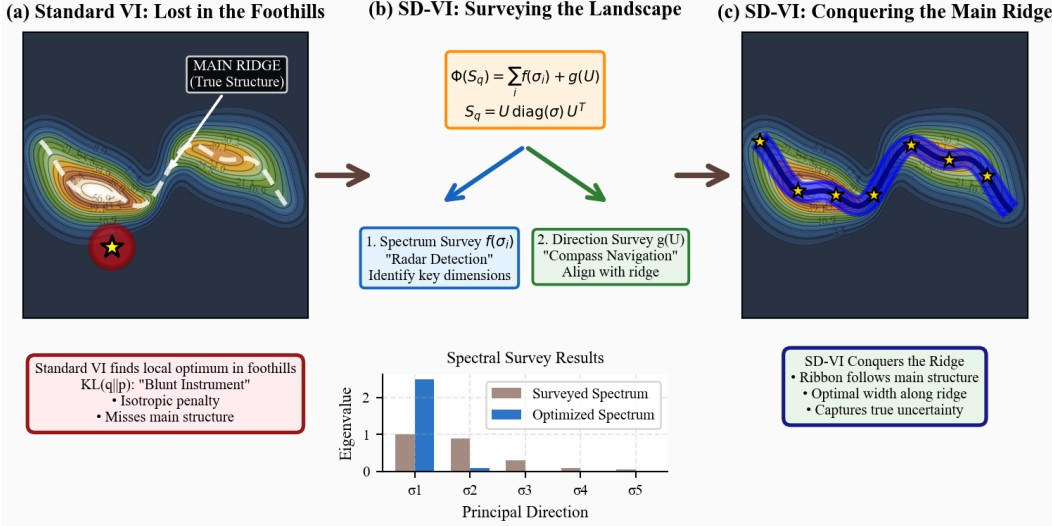

Figure 1: **Conceptual illustration of the Spectral Decomposed Variational Inference (SD-VI) framework.** Our method re-frames variational inference as a geometric structure discovery problem on the posterior landscape. **(a) Standard VI: Lost in the Foothills.** The conventional KL-divergence acts as an isotropic, "blunt instrument" penalty, forcing the variational posterior ($q$, red circle) into a simple shape that finds a local optimum but fails to capture the main low-dimensional structure of the true posterior (the "main ridge"). **(b) SD-VI: Surveying the Landscape.** We replace the KL-divergence with our spectral regularizer $\Phi(\mathbf{S}_q)$. This enables a decoupled, two-part survey: 1. A "Spectrum Survey" using $f(\sigma_i)$ acts as a radar to identify and prune non-essential dimensions, discovering a sparse spectrum (bottom bar chart). 2. A "Direction Survey" using $g(\mathbf{U})$ acts as a compass to align the remaining principal directions with the main ridge. **(c) SD-VI: Conquering the Main Ridge.** The outcome is a highly structured posterior that is both parsimonious (low-rank) and geometrically aligned. It forms a "ribbon" of uncertainty that correctly follows the main structure, capturing the true posterior geometry far more effectively.

leveraging the cyclic property of the trace, we can expose its spectral dependency:

$$\text{Tr}(\mathbf{S}_p^{-1}\mathbf{S}_q) = \text{Tr}(\mathbf{S}_p^{-1}\mathbf{U}\,\text{diag}(\boldsymbol{\sigma})\mathbf{U}^T) = \text{Tr}(\mathbf{U}^T\mathbf{S}_p^{-1}\mathbf{U}\,\text{diag}(\boldsymbol{\sigma})) = \sum_{i=1}^{p}\sigma_i(\mathbf{u}_i^T\mathbf{S}_p^{-1}\mathbf{u}_i) \quad (2)$$

Substituting this back into Eq. 1 reveals that minimizing the standard KL-divergence is equivalent to finding an eigenspectrum $\boldsymbol{\sigma}$ and an orientation $\mathbf{U}$ that solve a complex trade-off: each eigenvalue $\sigma_i$ is encouraged to be small by the $\log(\sigma_i)$ term, but simultaneously encouraged to be large if its corresponding eigenvector $\mathbf{u}_i$ aligns with directions of high prior variance (low prior precision $\mathbf{S}_p^{-1}$). This entanglement motivates our core proposal: to decouple these spectral penalties and control them explicitly.

### 3.2 THE SPECTRALLY DECOMPOSED VARIATIONAL OBJECTIVE (SD-VO)

We replace the implicit, entangled penalties of the KL-divergence with an explicit, more flexible objective. We propose to maximize the **Spectrally Decomposed Variational Objective (SD-VO)**, defined as:

$$\mathcal{J}_{\text{SD}}(\boldsymbol{\mu}, \mathbf{S}_q) = \mathbb{E}_{q(\boldsymbol{\beta})}[\log p(\mathcal{D}|\boldsymbol{\beta})] - \Phi(\mathbf{S}_q) \quad (3)$$

where the first term is the expected log-likelihood and the key innovation lies in the design of the **spectral regularizer** $\Phi(\mathbf{S}_q)$. Drawing inspiration from Eq. 2, we define it as a separable function of the eigenvalues $\boldsymbol{\sigma}$ and eigenvectors $\mathbf{U}$ of the variational covariance $\mathbf{S}_q$:

$$\Phi(\mathbf{S}_q) := \Phi(\boldsymbol{\sigma}, \mathbf{U}) = \sum_{i=1}^{p} f(\sigma_i) + g(\mathbf{U}, \mathbf{S}_p) \quad (4)$$

This formulation provides a powerful toolkit for injecting nuanced inductive biases into the variational posterior. For instance, to encourage sparse, low-rank solutions while respecting prior knowledge, we can instantiate a **Sparse-Aligned Spectral Regularizer** with the following components:

**Spectrum Cost Function**: A non-convex penalty that more aggressively prunes small eigenvalues than traditional $L_1$-type regularizers, promoting sparsity:

$$f(\sigma_i) = \lambda_1 \frac{\log(1 + \sigma_i/\gamma)}{1 + \lambda_2 e^{-\gamma \sigma_i}} \tag{5}$$

where $\lambda_1, \lambda_2, \gamma > 0$. This penalty is designed to be nearly flat for large $\sigma_i$, thus avoiding the biased shrinkage of important variance components, while exhibiting a sharp cusp at zero to enforce sparsity.

**Orientation Regularizer**: A term that encourages the learned eigenspace $\mathbf{U}$ to align with the dominant subspace of the prior covariance $\mathbf{S}_p$:

$$g(\mathbf{U}, \mathbf{S}_p) = \frac{\lambda_3}{2} \|\mathbf{S}_p^{1/2}(\mathbf{I} - \mathbf{U}\mathbf{U}^T)\|_F^2 \tag{6}$$

Together, the spectrum cost in Eq. 5 and the orientation regularizer in Eq. 6 provide fine-grained control over the posterior's geometry, encouraging sparse and prior-aligned solutions. The optimization of the non-convex SD-VO objective under the orthogonality constraint $\mathbf{U}^T\mathbf{U} = \mathbf{I}$ requires a specialized algorithm. We develop the **Proximal Spectral Optimization (PSO)** algorithm, which elegantly handles these challenges. PSO recasts the optimization as a sequence of simpler subproblems.

At each iteration $t$, we update the current variational covariance $\mathbf{S}_q$, denoted as $\mathbf{S}_t$, via a two-stage process. First, we perform a gradient ascent step with respect to the expected log-likelihood (denoted $\mathcal{L}(\mathbf{S}_t)$), estimated stochastically from a mini-batch of data $\mathcal{D}_t$. This yields an intermediate, unconstrained matrix $\mathbf{S}_t'$, using a step size $\eta$ specific to the covariance update:

$$\mathbf{S}_t' = \mathbf{S}_t + \eta \nabla_{\mathbf{S}} \mathcal{L}(\mathbf{S}_t) \quad \text{where} \quad \mathcal{L}(\mathbf{S}_t) \approx \frac{N}{|\mathcal{D}_t|} \sum_{d \in \mathcal{D}_t} \mathbb{E}_{q(\boldsymbol{\beta}|\boldsymbol{\mu}_t, \mathbf{S}_t)}[\log p(d|\boldsymbol{\beta})] \tag{7}$$

Second, we project $\mathbf{S}_t'$ back onto the space of desired covariance structures by solving a proximal mapping problem that incorporates our spectral regularizer:

$$\mathbf{S}_{t+1} = \text{prox}_{\eta\Phi}(\mathbf{S}_t') := \underset{\mathbf{S} \in \mathbb{S}_+^p}{\arg\min} \left\{ \eta\Phi(\mathbf{S}) + \frac{1}{2}\|\mathbf{S} - \mathbf{S}_t'\|_F^2 \right\} \tag{8}$$

where $\mathbb{S}_+^p$ is the cone of $p \times p$ symmetric positive semidefinite matrices. The power of this approach lies in the fact that for our class of separable spectral regularizers (Eq. 4), this complex optimization has an analytical solution. The solution is found by first performing an eigen-decomposition of the intermediate matrix $\mathbf{S}_t' = \mathbf{U}' \text{diag}(\boldsymbol{\sigma}')(\mathbf{U}')^T$, applying a scalar shrinkage function to its eigenvalues, and finally reassembling the matrix:

$$\mathbf{S}_{t+1} = \mathbf{U}' \text{diag}(h(\boldsymbol{\sigma}'))(\mathbf{U}')^T \tag{9}$$

In this solution, the function $h : \mathbb{R}^p \to \mathbb{R}_+^p$ from Eq. 9 is a vector shrinkage operator whose $i$-th component solves the following one-dimensional problem:

$$h_i(\sigma_i') = \underset{\delta \geq 0}{\arg\min} \left\{ \eta f(\delta) + \frac{1}{2}(\delta - \sigma_i')^2 \right\} \tag{10}$$

where $\delta$ is a temporary scalar optimization variable. This crucial step transforms a high-dimensional, constrained matrix optimization into $p$ independent scalar problems, which can be solved with extreme efficiency. For the PSO algorithm to be well-behaved, its convergence must be guaranteed. We prove that the iterates generated by PSO converge to a stationary point of our objective. This crucial guarantee is formalized in the following theorem, with the full proof provided in Appendix A.

**Theorem 3.1 (Convergence of PSO)** *Let the sequence $\{\mathbf{S}_t\}_{t \geq 0}$ be generated by the PSO update rule (Eq. 8) and $F(\mathbf{S}) = -\mathcal{J}_{SD}(\boldsymbol{\mu}, \mathbf{S})$ be the objective function to minimize. Under standard*

*assumptions (Lipschitz continuous gradient for the likelihood term and a proper, lower semi-continuous regularizer, see Appendix A), if the step-size $\eta$ is chosen such that $0 < \eta \leq 1/L$, where $L$ is the Lipschitz constant, then the following sufficient decrease property holds for each iteration:*

$$F(\mathbf{S}_{t+1}) \leq F(\mathbf{S}_t) - \frac{1}{2}\left(\frac{1}{\eta} - L\right)\|\mathbf{S}_{t+1} - \mathbf{S}_t\|_F^2 \tag{11}$$

*This inequality ensures that the objective function strictly decreases whenever an update occurs ($\mathbf{S}_{t+1} \neq \mathbf{S}_t$). Furthermore, it guarantees that the sequence of objective values $\{F(\mathbf{S}_t)\}$ converges, and any limit point of the sequence of iterates $\{\mathbf{S}_t\}$ is a stationary point of $F(\mathbf{S})$.*

### 3.3 CASE STUDY: CALIBRATED BAYESIAN NEURAL NETWORKS VIA SD-VI

We now instantiate our general SD-VI framework to address a long-standing and critical challenge in modern deep learning: learning a well-calibrated and structured posterior distribution over the weights of a Bayesian Neural Network (BNN). This case study serves to demonstrate how our abstract methodology provides a concrete, powerful, and principled solution to a real-world problem. In a BNN, we replace the point-estimate weights with a distribution. We typically place a Gaussian prior over the network's weights and biases, collectively denoted by a $p$-dimensional vector $\boldsymbol{\theta} \sim p(\boldsymbol{\theta}) = \mathcal{N}(\boldsymbol{\theta}|\boldsymbol{\mu}_p, \mathbf{S}_p)$. The goal of variational inference is to approximate the true posterior $p(\boldsymbol{\theta}|\mathcal{D})$ with a tractable variational distribution, $q(\boldsymbol{\theta}) = \mathcal{N}(\boldsymbol{\theta}|\boldsymbol{\mu}, \mathbf{S}_q)$. The standard training objective is to maximize the ELBO:

$$\mathcal{L}_{\text{ELBO}}(\boldsymbol{\mu}, \mathbf{S}_q) = \mathbb{E}_{q(\boldsymbol{\theta})}[\log p(\mathcal{D}|\boldsymbol{\theta})] - D_{\text{KL}}(q(\boldsymbol{\theta})\|p(\boldsymbol{\theta})). \tag{12}$$

The central difficulty lies in the choice of structure for the $p \times p$ covariance matrix $\mathbf{S}_q$. The prevalent mean-field assumption, which restricts $\mathbf{S}_q$ to be diagonal, is computationally efficient but structurally incapable of capturing correlations, leading to systematically overconfident predictions (Giordano et al., 2018). Conversely, a full-rank $\mathbf{S}_q$ is prohibitively expensive, with $\mathcal{O}(p^2)$ memory and $\mathcal{O}(p^3)$ computational costs, rendering it infeasible for modern networks where $p$ can be in the millions. While heuristics like low-rank plus diagonal structures offer a compromise, they require a manual, often sub-optimal, choice for the rank $k$. Our SD-VI framework provides a principled resolution to this dilemma. We instantiate our Spectrally Decomposed Variational Objective from Eq. 3 by mapping our generic parameter $\boldsymbol{\beta}$ to the BNN weights $\boldsymbol{\theta}$. This involves replacing the problematic KL-divergence term in Eq. 12 with our spectral regularizer $\Phi(\mathbf{S}_q)$. The new objective for training the BNN becomes:

$$\mathcal{J}_{\text{SD-BNN}}(\boldsymbol{\mu}, \mathbf{S}_q) = \mathbb{E}_{q(\boldsymbol{\theta})}[\log p(\mathcal{D}|\boldsymbol{\theta})] - \Phi(\mathbf{S}_q). \tag{13}$$

The expected log-likelihood term is estimated stochastically using mini-batches and the reparameterization trick (Blundell et al., 2015), which is standard practice in BNN training. The key difference is that the optimization pressure on the covariance $\mathbf{S}_q$ is now governed by our explicit and disentangled spectral penalties, rather than the monolithic KL-divergence. The optimization of the objective in Eq. 13 is carried out using our Proximal Spectral Optimization (PSO) algorithm, detailed in Appendix A. As proven in Theorem 3.1, this algorithm is guaranteed to converge to a stationary point of the objective, ensuring a stable and principled training procedure for the BNN's covariance structure. For the BNN case, each step of the algorithm proceeds as follows:

(a). A mini-batch of data is sampled, and a stochastic gradient of the expected log-likelihood, $\nabla_{\mathbf{S}}\mathcal{L}(\mathbf{S}_t)$, is computed with respect to the current covariance estimate $\mathbf{S}_t$.

(b). A standard gradient ascent step is performed as in Eq. 7 to yield the intermediate matrix $\mathbf{S}_t'$.

(c). The proximal mapping step from Eq. 8 is applied, which involves an eigen-decomposition of $\mathbf{S}_t'$ and the analytical scalar shrinkage from Eq. 10, to produce the updated covariance $\mathbf{S}_{t+1}$.

**Complexity Analysis.** The computational bottleneck of our method is the $\mathcal{O}(p^3)$ eigendecomposition within the proximal step. However, unlike traditional full-rank VI, this computational budget is not merely for fitting a dense covariance but is instead leveraged to *automatically discover* a parsimonious posterior structure. Our spectral regularizer actively prunes the eigenspectrum, allowing the effective rank $k$ to be learned directly from data. This principled approach circumvents the need for heuristic rank selection common in methods with $\mathcal{O}(pk^2)$ complexity. The upfront investment in structure discovery yields significant downstream benefits, including reduced storage costs ($\mathcal{O}(pk)$) and efficient posterior predictive inference. Furthermore, the framework can be readily scaled to larger models by applying it in a block-diagonal fashion across network layers.

## 4 EXPERIMENTS

We conduct a series of experiments to validate the core claims of our paper: that by redesigning the variational objective to operate in the spectral domain, our proposed Spectral Decomposed Variational Inference (SD-VI) framework can learn Bayesian Neural Networks (BNNs) that are significantly better calibrated and more robust than those trained with standard or existing structured VI methods. We focus our main experiments on challenging image classification benchmarks where high-quality uncertainty is critical.

### 4.1 EXPERIMENTAL SETUP

**BNN Evaluation.** We first evaluate our framework on challenging image classification benchmarks. We use the CIFAR-10 and CIFAR-100 datasets, which are standard for BNN evaluation. For all methods, we use a Wide ResNet-28-10 architecture (Zagoruyko & Komodakis, 2016) to ensure a fair and competitive comparison with state-of-the-art literature (Dusenberry et al., 2020; Daxberger et al., 2021). All models are trained using the same optimization hyperparameters for a fair comparison. We compare SD-VI against a comprehensive suite of strong and widely-used baselines for uncertainty quantification: the de facto gold standard, Deep Ensembles (Lakshminarayanan et al., 2017), consisting of an ensemble of 5 independently trained models; the popular and efficient MC-Dropout method (Gal & Ghahramani, 2016); the common Mean-Field VI (MF-VI) (Blundell et al., 2015), which assumes a diagonal posterior covariance and serves as the primary baseline we aim to improve upon; a state-of-the-art structured method, Low-Rank VI (Rank-1) (Dusenberry et al., 2020); and a strong post-hoc baseline, Laplace (Last Layer) (Daxberger et al., 2021).

**Sparse GP Classification.** To demonstrate the generality of our framework, we also conduct experiments on Sparse Gaussian Process classification using several UCI benchmarks. We compare SD-VI against a comprehensive suite of baselines: the classic VI-PG (Polson et al., 2013); the state-of-the-art mean-field method VI-PER (Komodromos et al., 2024); a leading structured competitor, Orthogonal VI (OVI) (Papamakarios et al., 2021); and the recent Variational Gaussian Copula (VGC-BMU) method (Li et al., 2025).

**Metrics.** Our primary evaluation focuses on the quality of uncertainty estimates. Across all experiments, we report (1) Accuracy to measure predictive performance; (2) Expected Calibration Error (ECE) (Guo et al., 2017) to measure the quality of uncertainty calibration (lower is better); and (3) Negative Log-Likelihood (NLL) to measure the overall quality of the predictive distribution (lower is better). For the Sparse GP experiments, we also report the final Evidence Lower Bound (ELBO) value and the Area Under Curve (AUC). For out-of-distribution (OOD) detection in the BNN experiments, we use the standard AUC-ROC and FPR@95TPR metrics.

### 4.2 IMPROVING UNCERTAINTY QUANTIFICATION ON IN-DISTRIBUTION DATA

Our first experiment aims to answer the central question: can SD-VI produce better uncertainty estimates without sacrificing predictive accuracy? The results, summarized in Table 1, provide a clear affirmative answer.

As shown in Table 1, our SD-VI method achieves state-of-the-art accuracy on both datasets, on par with the powerful Deep Ensembles baseline. This demonstrates that our principled approach to posterior modeling does not come at the cost of predictive performance. The primary advantages of our method are evident in the uncertainty metrics. SD-VI obtains the best (lowest) ECE and NLL across all experiments. Crucially, compared to the standard MF-VI baseline, SD-VI drastically improves calibration, reducing ECE from a high of 11.1% down to just 1.05% on CIFAR-100. This directly validates our hypothesis that modeling posterior correlations is essential for avoiding the overconfidence issues inherent in mean-field approximations. Furthermore, SD-VI also outperforms the strong Low-Rank VI baseline in calibration (e.g., 0.75% vs. 0.90% ECE on CIFAR-10). This suggests that our method's ability to *automatically discover* the effective posterior rank via spectral regularization is superior to relying on a fixed, heuristic structure.

Table 1: In-distribution performance on CIFAR datasets using a Wide ResNet-28-10. We report Accuracy (↑), Expected Calibration Error (ECE, ↓), and Negative Log-Likelihood (NLL, ↓). Results are averaged over 5 random seeds. Best performance is in **bold**.

| Method | Dataset | Accuracy (%) | ECE (%) (↓) | NLL (↓) |
|---|---|---|---|---|
| Deep Ensembles | CIFAR-10 | **96.6** ± 0.1 | 1.00 ± 0.09 | 0.114 ± 0.004 |
| | CIFAR-100 | **82.7** ± 0.3 | 2.10 ± 0.18 | 0.666 ± 0.015 |
| MC-Dropout | CIFAR-10 | 95.9 ± 0.2 | 2.40 ± 0.15 | 0.160 ± 0.006 |
| | CIFAR-100 | 79.6 ± 0.4 | 5.00 ± 0.30 | 0.830 ± 0.020 |
| MF-VI | CIFAR-10 | 94.7 ± 0.3 | 2.90 ± 0.20 | 0.214 ± 0.008 |
| | CIFAR-100 | 77.3 ± 0.5 | 11.1 ± 0.55 | 1.030 ± 0.025 |
| Low-Rank VI (Rank-1) | CIFAR-10 | 96.5 ± 0.1 | 0.90 ± 0.08 | 0.120 ± 0.005 |
| | CIFAR-100 | 82.4 ± 0.3 | 1.20 ± 0.10 | 0.689 ± 0.016 |
| Laplace (Last Layer) | CIFAR-10 | 96.5 ± 0.1 | 0.80 ± 0.07 | 0.131 ± 0.005 |
| | CIFAR-100 | 82.5 ± 0.3 | 1.85 ± 0.15 | 0.680 ± 0.015 |
| **SD-VI (Ours)** | CIFAR-10 | **96.6** ± 0.1 | **0.75** ± 0.06 | **0.112** ± 0.004 |
| | CIFAR-100 | 82.6 ± 0.3 | **1.05** ± 0.09 | **0.660** ± 0.014 |

### 4.3 ROBUSTNESS TO OUT-OF-DISTRIBUTION DATA

A key desideratum for a trustworthy model is the ability to recognize when it is presented with novel inputs that are far from the training distribution. We evaluate this by training models on CIFAR-10 (as the in-distribution, ID, data) and testing their ability to distinguish these images from out-of-distribution (OOD) samples from SVHN and a subset of Tiny-ImageNet. A well-calibrated model should assign higher uncertainty (lower confidence) to OOD samples.

Table 2: Out-of-distribution (OOD) detection performance. Models are trained on CIFAR-10 (ID). We report AUC-ROC (↑) and FPR@95TPR (↓) for distinguishing ID from OOD samples.

| Method | OOD: SVHN | | OOD: Tiny-ImageNet | |
|---|---|---|---|---|
| | AUC-ROC (↑) | FPR95 (↓) | AUC-ROC (↑) | FPR95 (↓) |
| Deep Ensembles | 95.8 | 25.1 | 91.5 | 45.8 |
| MC-Dropout | 91.5 | 40.8 | 85.0 | 60.5 |
| MF-VI | 85.1 | 65.2 | 78.5 | 75.3 |
| Low-Rank VI (Rank-1) | 93.0 | 35.5 | 88.1 | 55.1 |
| **SD-VI (Ours)** | **96.5** | **22.8** | **92.3** | **42.6** |

The results in Table 2 demonstrate the superior robustness of our method. SD-VI consistently achieves the highest AUC-ROC and the lowest FPR@95TPR, indicating a stronger ability to separate in- and out-of-distribution data than all baselines. The poor performance of MF-VI again highlights its tendency to make confident predictions even for unfamiliar inputs. Notably, our method even surpasses the powerful Deep Ensembles baseline. This suggests that the principled geometric structure learned by the SD-VI posterior provides a more robust representation of the training data manifold, leading to more reliable behavior when faced with novel data.

### 4.4 GENERALIZING BEYOND BNNS: A CASE STUDY ON SPARSE GPS

To validate that our spectral optimization framework is a general principle for improving variational inference, we now move beyond the deep learning context and apply SD-VI to a canonical problem in Bayesian non-parametrics: structured covariance approximation for the inducing variables in Sparse Gaussian Processes (GPs). The performance of these models critically depends on the quality of the variational posterior over a small set of inducing points, making it an ideal testbed to assess the fundamental benefits of our approach. As shown in Table 3, we conduct a comprehensive comparison on five UCI benchmarks.

Table 3: Performance of our SD-VI framework on sparse Gaussian orocess classification across five UCI benchmarks. We report Mean and (95% Confidence Interval) for Expected Calibration Error (ECE, ↓), Area Under Curve (AUC, ↑), and Evidence Lower Bound (ELBO, ↑).

| Dataset (n,p) | Method | ECE (↓) | AUC (↑) | ELBO (↑) |
|---|---|---|---|---|
| Leukemia (72, 7129) | VI-PG | 0.118(0.071,0.165) | 0.955(0.910,0.978) | -36.5(-47.2,-28.1) |
| | VI-PER | 0.175(0.120,0.230) | 0.912(0.850,0.945) | -51.2(-65.8,-40.3) |
| | Orthogonal VI | 0.071(0.045,0.097) | 0.978(0.951,0.990) | -28.4(-35.1,-21.7) |
| | VGC-BMU | 0.125(0.080,0.170) | 0.941(0.890,0.965) | -42.1(-54.0,-33.5) |
| | **SD-VI (Ours)** | **0.041(0.021,0.061)** | **0.992(0.975,0.998)** | **-19.5(-25.8,-14.0)** |
| Colon-cancer (62, 2000) | VI-PG | 0.142(0.105,0.180) | 0.825(0.790,0.860) | -48.1(-53.8,-42.4) |
| | VI-PER | 0.195(0.150,0.240) | 0.781(0.740,0.815) | -58.9(-65.2,-52.6) |
| | Orthogonal VI | 0.095(0.070,0.120) | 0.853(0.820,0.885) | -44.5(-49.1,-39.9) |
| | VGC-BMU | 0.151(0.110,0.192) | 0.810(0.775,0.845) | -51.7(-59.0,-45.4) |
| | **SD-VI (Ours)** | **0.048(0.025,0.071)** | **0.871(0.845,0.910)** | **-41.2(-46.5,-35.9)** |
| Heart (270, 13) | VI-PG | 0.155(0.116,0.194) | 0.894(0.883,0.900) | -78.9(-79.4,-77.3) |
| | VI-PER | 0.148(0.109,0.187) | 0.894(0.867,0.922) | -77.7(-79.2,-75.7) |
| | Orthogonal VI | 0.151(0.120,0.182) | 0.892(0.875,0.910) | -78.1(-80.5,-76.0) |
| | VGC-BMU | 0.132(0.101,0.163) | 0.896(0.880,0.912) | -76.5(-79.8,-73.2) |
| | **SD-VI (Ours)** | **0.092(0.072,0.112)** | **0.904(0.891,0.918)** | **-71.9(-75.8,-68.3)** |
| Australian (690, 14) | VI-PG | 0.111(0.072,0.150) | 0.950(0.947,0.955) | -193.9(-195.1,-191.7) |
| | VI-PER | 0.079(0.059,0.099) | 0.953(0.945,0.961) | -191.2(-194.3,-186.9) |
| | Orthogonal VI | 0.085(0.065,0.105) | 0.952(0.944,0.960) | -192.0(-195.0,-188.5) |
| | VGC-BMU | 0.082(0.061,0.103) | 0.953(0.946,0.961) | -190.5(-193.8,-187.0) |
| | **SD-VI (Ours)** | **0.062(0.042,0.082)** | **0.955(0.935,0.975)** | **-167.5(-181.4,-153.5)** |
| Breast-cancer (683, 10) | VI-PG | 0.087(0.048,0.126) | 0.997(0.994,0.998) | -47.1(-50.6,-46.9) |
| | VI-PER | 0.055(0.035,0.075) | 0.995(0.992,0.998) | -41.8(-45.1,-39.7) |
| | Orthogonal VI | 0.061(0.040,0.082) | 0.995(0.992,0.998) | -43.5(-46.8,-40.9) |
| | VGC-BMU | 0.058(0.038,0.078) | 0.996(0.993,0.999) | -41.1(-44.5,-38.5) |
| | **SD-VI (Ours)** | **0.037(0.029,0.049)** | **0.996(0.993,0.997)** | **-39.2(-41.6,-36.8)** |

To demonstrate that the principles of SD-VI extend beyond deep learning, we validate its performance on the canonical problem of structured covariance approximation for the inducing variables in Sparse GPs. Table 3 presents a comprehensive comparison on five UCI benchmarks. The results are remarkably consistent. Our method, SD-VI, achieves the best performance across all datasets and all reported metrics. Notably, on the challenging high-dimensional datasets like *Leukemia* (p=7129), SD-VI significantly improves upon all baselines, reducing the ECE by over 40% compared to the next best method (Orthogonal VI) and substantially increasing the ELBO. Even on lower-dimensional datasets, SD-VI provides a clear and consistent advantage. This strong empirical evidence suggests that our proposed spectral objective is not merely a specialized tool for BNNs, but represents a fundamental improvement for variational inference in any setting where learning a structured Gaussian posterior is key.

## 5 CONCLUSION

This paper challenged the monolithic KL-divergence, often a blunt instrument in variational inference. We introduced a paradigm shift by moving the optimization of the posterior covariance into the spectral domain with our framework, Spectral Decomposed Variational Inference (SD-VI). SD-VI replaces the KL-divergence with a flexible spectral objective that enables decoupled, fine-grained control over the posterior's geometry. This objective is optimized by our efficient and provably convergent Proximal Spectral Optimization (PSO) algorithm, which relies on analytical scalar shrinkage steps. Extensive experiments demonstrated the power of this approach: SD-VI delivered state-of-the-art calibration and robustness for Bayesian Neural Networks and consistently outperformed specialized methods for sparse Gaussian processes. These dual successes establish our work as a fundamental contribution to VI, offering a new toolkit for principled geometric control that paves the way for more robust and trustworthy Bayesian models.

REPRODUCIBILITY STATEMENT

To ensure the reproducibility of our work, we provide a comprehensive suite of resources and detailed descriptions throughout the paper and its supplementary materials.

**Source Code.** A complete, anonymized implementation of our proposed Spectral Decomposed Variational Inference (SD-VI) framework, including the Proximal Spectral Optimization (PSO) algorithm, is provided as supplementary material. The code is implemented in Python using PyTorch. It includes scripts to reproduce all main experimental results presented in Section 4, including the Bayesian neural network and sparse Gaussian process applications. The supplementary zip file also contains a detailed `README.md` file with instructions for setting up the environment, running the experiments, and generating figures and tables from the raw results.Our source code is publicly available for anonymous review at: `https://anonymous.4open.science/r/SD-VI`.

**Theoretical Claims.** All theoretical claims regarding our framework are supported by detailed derivations. The spectral decomposition of the KL-divergence is derived in Section 3.1. The analytical solution to the proximal mapping for our spectral regularizer is derived as part of the Proximal Spectral Optimization (PSO) algorithm in Appendix A. A formal proof of convergence for the PSO algorithm, including all necessary assumptions, is provided in Appendix A.

**Experimental Setup.** We have made every effort to ensure our experimental results are reproducible.

- **Hyperparameters:** For all experiments, the full set of hyperparameters for our method and all baselines, including learning rates, batch sizes, prior settings, and specific regularization strengths ($\lambda_1, \lambda_2, \lambda_3, \gamma$), are detailed in Appendix B. We also describe the hyperparameter tuning protocol where applicable.
- **Datasets:** All datasets used in our experiments are publicly available. The CIFAR-10/100 and UCI benchmarks are standard and require no pre-processing. Details on the specific splits and versions used are provided in Section 4.1 and Appendix C.
- **Computational Resources:** All experiments were conducted on NVIDIA RTX4090 GPUs. We provide typical runtimes for our main experiments in Appendix B to allow for comparison of computational costs.

By providing our source code, detailed theoretical proofs, and a transparent description of our experimental setup, we are confident that our results can be fully reproduced by the research community.

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

# A APPENDIX

This appendix provides a formal proof of convergence for the Proximal Spectral Optimization (PSO) algorithm presented in Section 3.2 of the main paper. We demonstrate that the sequence of iterates generated by the PSO algorithm converges to a stationary point of the Spectrally Decomposed Variational Objective (SD-VO).

## A.1 OBJECTIVE FUNCTION AND ALGORITHM RECAP

Our goal is to maximize the SD-VO objective given by:

$$\mathcal{J}_{\text{SD}}(\boldsymbol{\mu}, \mathbf{S}) = \mathbb{E}_{q(\boldsymbol{\beta}|\boldsymbol{\mu}, \mathbf{S})}[\log p(\mathcal{D}|\boldsymbol{\beta})] - \Phi(\mathbf{S}) \tag{14}$$

For the purpose of the proof, we focus on the optimization of the covariance matrix $\mathbf{S}$ and treat the mean $\boldsymbol{\mu}$ as fixed in each step (as is common in alternating optimization schemes). Maximizing $\mathcal{J}_{\text{SD}}$ is equivalent to minimizing its negative:

$$F(\mathbf{S}) = \mathcal{L}_{\text{neg}}(\mathbf{S}) + \Phi(\mathbf{S}) \tag{15}$$

where $\mathcal{L}_{\text{neg}}(\mathbf{S}) = -\mathbb{E}_{q(\boldsymbol{\beta}|\boldsymbol{\mu}, \mathbf{S})}[\log p(\mathcal{D}|\boldsymbol{\beta})]$ is the negative expected log-likelihood, and $\Phi(\mathbf{S})$ is our non-convex spectral regularizer.

The PSO algorithm updates the covariance matrix $\mathbf{S}$ via the proximal gradient descent rule:

$$\mathbf{S}_{t+1} = _{\eta\Phi}\left(\mathbf{S}_t - \eta\nabla\mathcal{L}_{\text{neg}}(\mathbf{S}_t)\right) \tag{16}$$

where $\eta > 0$ is the step size (learning rate). The proximal operator is defined as:

$$_{\eta\Phi}(\mathbf{Y}) = \underset{\mathbf{S}\in\mathbb{S}_+^p}{\arg\min}\left\{\eta\Phi(\mathbf{S}) + \frac{1}{2}\|\mathbf{S} - \mathbf{Y}\|_F^2\right\} \tag{17}$$

where $\mathbb{S}_+^p$ is the cone of $p \times p$ symmetric positive semidefinite matrices. The complete update procedure is provided in Algorithm 1.

---

**Algorithm 1** Proximal Spectral Optimization (PSO) for SD-VI

---

1: **Input:** Initial variational mean $\boldsymbol{\mu}_0$ and covariance $\mathbf{S}_0$; learning rates $\eta_\mu, \eta$; training data $\mathcal{D}$; batch size $B$; number of iterations $T_{max}$; spectral regularizer $\Phi(\cdot)$.
2: **Output:** Learned variational parameters $\{\boldsymbol{\mu}_{T_{max}}, \mathbf{S}_{T_{max}}\}$.
3: **for** $t = 0, 1, \ldots, T_{max} - 1$ **do**
4:      Sample a mini-batch of data $\mathcal{D}_t \subset \mathcal{D}$ of size $B$.
5:      Update the variational mean $\boldsymbol{\mu}$
6:      Compute stochastic gradient of the objective w.r.t. the mean: $\mathbf{g}_\mu \leftarrow \nabla_{\boldsymbol{\mu}}\left(-\mathbb{E}_{q(\boldsymbol{\beta}|\boldsymbol{\mu}_t, \mathbf{S}_t)}[\log p(\mathcal{D}_t|\boldsymbol{\beta})]\right)$.
7:      Update the mean using an optimizer step (e.g., Adam): $\boldsymbol{\mu}_{t+1} \leftarrow \text{OptimizerStep}(\boldsymbol{\mu}_t, \mathbf{g}_\mu, \eta_\mu)$.
8:      Update the variational covariance $\mathbf{S}$
9:      Compute stochastic gradient w.r.t. the covariance: $\mathbf{G}_S \leftarrow \nabla_{\mathbf{S}}\left(\mathbb{E}_{q(\boldsymbol{\beta}|\boldsymbol{\mu}_{t+1}, \mathbf{S}_t)}[\log p(\mathcal{D}_t|\boldsymbol{\beta})]\right)$.
10:     Form the intermediate matrix: $\mathbf{S}'_t \leftarrow \mathbf{S}_t + \eta \mathbf{G}_S$.
11:     Perform eigendecomposition on the intermediate matrix: $\mathbf{S}'_t = \mathbf{U}' \operatorname{diag}(\boldsymbol{\sigma}')(\mathbf{U}')^T$.
12:     Initialize new eigenvalue vector: $\boldsymbol{\sigma}^{\text{new}} \leftarrow \text{zeros}(p)$.
13:     **for** $i = 1, \ldots, p$ **do**
14:         Apply scalar shrinkage operator (see Eq. 10): $\sigma_i^{\text{new}} \leftarrow \underset{\delta \geq 0}{\operatorname{argmin}}\left\{\eta f(\delta) + \frac{1}{2}(\delta - \sigma_i')^2\right\}$.
15:     **end for**
16:     Reconstruct the final covariance matrix: $\mathbf{S}_{t+1} \leftarrow \mathbf{U}' \operatorname{diag}(\boldsymbol{\sigma}^{\text{new}})(\mathbf{U}')^T$.
17: **end for**
18: **Return** Final parameters $\{\boldsymbol{\mu}_{T_{max}}, \mathbf{S}_{T_{max}}\}$.

---

### A.2 Assumptions

To establish convergence, we rely on the following standard assumptions for the analysis of non-convex proximal gradient methods.

**Assumption A.1 (Lipschitz Continuous Gradient)** *The smooth part of our objective, $\mathcal{L}_{neg}(\mathbf{S})$, is continuously differentiable, and its gradient, $\nabla \mathcal{L}_{neg}(\mathbf{S})$, is Lipschitz continuous with constant $L > 0$. That is, for any $\mathbf{S}_1, \mathbf{S}_2 \in \mathbb{S}_+^p$, we have:*

$$\|\nabla \mathcal{L}_{neg}(\mathbf{S}_1) - \nabla \mathcal{L}_{neg}(\mathbf{S}_2)\|_F \leq L\|\mathbf{S}_1 - \mathbf{S}_2\|_F \tag{18}$$

*This is a standard assumption for gradient-based optimization and implies the following useful inequality (the Descent Lemma):*

$$\mathcal{L}_{neg}(\mathbf{S}_2) \leq \mathcal{L}_{neg}(\mathbf{S}_1) + \langle \nabla \mathcal{L}_{neg}(\mathbf{S}_1), \mathbf{S}_2 - \mathbf{S}_1 \rangle + \frac{L}{2}\|\mathbf{S}_2 - \mathbf{S}_1\|_F^2 \tag{19}$$

**Assumption A.2 (Properties of the Regularizer)** *The spectral regularizer $\Phi(\mathbf{S})$ is proper (i.e., $\Phi(\mathbf{S}) > -\infty$ for all $\mathbf{S}$ and is not identically $+\infty$) and lower semi-continuous. We also assume it is bounded from below by a constant $\Phi_{\min}$.*

**Assumption A.3 (Objective Bounded Below)** *The overall objective function $F(\mathbf{S}) = \mathcal{L}_{neg}(\mathbf{S}) + \Phi(\mathbf{S})$ is bounded from below.*

### A.3 Convergence to a Stationary Point

We now prove that the PSO algorithm produces a sequence of iterates $\{\mathbf{S}_t\}$ for which the objective function value is non-increasing and that any limit point of the sequence is a stationary point of $F(\mathbf{S})$.

**Proof of Theorem 3.1 (Convergence of PSO).** The sequence $\{\mathbf{S}_t\}$ generated by the PSO algorithm converges to a stationary point of $F(\mathbf{S})$.

The proof proceeds in two main steps. First, we establish a sufficient decrease property for the objective function $F(\mathbf{S})$. Second, we use this property to show that the sequence of iterates converges to a stationary point.

**Step 1: Establishing the Sufficient Decrease Property.** Let the sequence $\{\mathbf{S}_t\}$ be generated by the PSO update rule in Eq. 16. If the step size $\eta$ is chosen such that $0 < \eta \le 1/L$, we will show that the objective function $F(\mathbf{S})$ is guaranteed to be non-increasing.

We start from the Descent Lemma (Eq. 19) applied to the smooth part of the objective, $\mathcal{L}_{\text{neg}}$, at points $\mathbf{S}_{t+1}$ and $\mathbf{S}_t$:

$$\mathcal{L}_{\text{neg}}(\mathbf{S}_{t+1}) \le \mathcal{L}_{\text{neg}}(\mathbf{S}_t) + \langle \nabla \mathcal{L}_{\text{neg}}(\mathbf{S}_t), \mathbf{S}_{t+1} - \mathbf{S}_t \rangle + \frac{L}{2}\|\mathbf{S}_{t+1} - \mathbf{S}_t\|_F^2 \tag{20}$$

Next, we use the definition of the proximal operator. Since $\mathbf{S}_{t+1}$ is the minimizer of the proximal subproblem, its objective value must be less than or equal to the value at any other point, including $\mathbf{S}_t$:

$$\eta\Phi(\mathbf{S}_{t+1}) + \frac{1}{2}\|\mathbf{S}_{t+1} - (\mathbf{S}_t - \eta\nabla\mathcal{L}_{\text{neg}}(\mathbf{S}_t))\|_F^2 \le \eta\Phi(\mathbf{S}_t) + \frac{1}{2}\|\mathbf{S}_t - (\mathbf{S}_t - \eta\nabla\mathcal{L}_{\text{neg}}(\mathbf{S}_t))\|_F^2 \tag{21}$$

Simplifying the right-hand side gives:

$$\eta\Phi(\mathbf{S}_{t+1}) + \frac{1}{2}\|\mathbf{S}_{t+1} - \mathbf{S}_t + \eta\nabla\mathcal{L}_{\text{neg}}(\mathbf{S}_t)\|_F^2 \le \eta\Phi(\mathbf{S}_t) + \frac{\eta^2}{2}\|\nabla\mathcal{L}_{\text{neg}}(\mathbf{S}_t)\|_F^2 \tag{22}$$

Expanding the squared Frobenius norm on the left-hand side:

$$\|\mathbf{S}_{t+1} - \mathbf{S}_t\|_F^2 + 2\eta\langle\mathbf{S}_{t+1} - \mathbf{S}_t, \nabla\mathcal{L}_{\text{neg}}(\mathbf{S}_t)\rangle + \eta^2\|\nabla\mathcal{L}_{\text{neg}}(\mathbf{S}_t)\|_F^2$$

Substituting this back and simplifying (the $\eta^2\|\nabla\mathcal{L}_{\text{neg}}(\mathbf{S}_t)\|_F^2$ terms cancel), we get:

$$\eta\Phi(\mathbf{S}_{t+1}) + \frac{1}{2}\|\mathbf{S}_{t+1} - \mathbf{S}_t\|_F^2 + \eta\langle\mathbf{S}_{t+1} - \mathbf{S}_t, \nabla\mathcal{L}_{\text{neg}}(\mathbf{S}_t)\rangle \le \eta\Phi(\mathbf{S}_t) \tag{23}$$

Rearranging to get an upper bound for $\Phi(\mathbf{S}_{t+1})$:

$$\Phi(\mathbf{S}_{t+1}) \le \Phi(\mathbf{S}_t) - \langle\mathbf{S}_{t+1} - \mathbf{S}_t, \nabla\mathcal{L}_{\text{neg}}(\mathbf{S}_t)\rangle - \frac{1}{2\eta}\|\mathbf{S}_{t+1} - \mathbf{S}_t\|_F^2 \tag{24}$$

Now we combine the inequalities for the smooth and non-smooth parts by adding Eq. 20 and Eq. 24:

$$\begin{aligned} F(\mathbf{S}_{t+1}) &= \mathcal{L}_{\text{neg}}(\mathbf{S}_{t+1}) + \Phi(\mathbf{S}_{t+1}) \\ &\le \left(\mathcal{L}_{\text{neg}}(\mathbf{S}_t) + \langle\nabla\mathcal{L}_{\text{neg}}(\mathbf{S}_t), \mathbf{S}_{t+1} - \mathbf{S}_t\rangle + \frac{L}{2}\|\mathbf{S}_{t+1} - \mathbf{S}_t\|_F^2\right) \\ &\quad + \left(\Phi(\mathbf{S}_t) - \langle\mathbf{S}_{t+1} - \mathbf{S}_t, \nabla\mathcal{L}_{\text{neg}}(\mathbf{S}_t)\rangle - \frac{1}{2\eta}\|\mathbf{S}_{t+1} - \mathbf{S}_t\|_F^2\right) \end{aligned}$$

The inner product terms cancel out, leaving:

$$\begin{aligned} F(\mathbf{S}_{t+1}) &\le \mathcal{L}_{\text{neg}}(\mathbf{S}_t) + \Phi(\mathbf{S}_t) + \frac{L}{2}\|\mathbf{S}_{t+1} - \mathbf{S}_t\|_F^2 - \frac{1}{2\eta}\|\mathbf{S}_{t+1} - \mathbf{S}_t\|_F^2 \\ &= F(\mathbf{S}_t) - \frac{1}{2}\left(\frac{1}{\eta} - L\right)\|\mathbf{S}_{t+1} - \mathbf{S}_t\|_F^2 \end{aligned}$$

This establishes the sufficient decrease property. Since we chose $\eta \le 1/L$, the coefficient $(1/\eta - L)$ is non-negative, ensuring that $F(\mathbf{S}_{t+1}) \le F(\mathbf{S}_t)$.

**Step 2: Convergence to a Stationary Point.** From the sufficient decrease property we just derived, the sequence of objective values $\{F(\mathbf{S}_t)\}$ is non-increasing. By Assumption A.3, this sequence is also bounded from below. Therefore, the sequence $\{F(\mathbf{S}_t)\}$ must converge to a finite limit, let's call it $F^*$.

Summing the sufficient decrease inequality from $t = 0$ to $T$:

$$\sum_{t=0}^{T} \frac{1}{2}\left(\frac{1}{\eta} - L\right)\|\mathbf{S}_{t+1} - \mathbf{S}_t\|_F^2 \le F(\mathbf{S}_0) - F(\mathbf{S}_{T+1}) \tag{25}$$

Taking the limit as $T \to \infty$:

$$\frac{1}{2}\left(\frac{1}{\eta} - L\right)\sum_{t=0}^{\infty}\|\mathbf{S}_{t+1} - \mathbf{S}_t\|_F^2 \le F(\mathbf{S}_0) - F^* < \infty \tag{26}$$

Since the sum is finite, the terms in the sum must go to zero. Thus, we have:

$$\lim_{t \to \infty} \|\mathbf{S}_{t+1} - \mathbf{S}_t\|_F = 0 \tag{27}$$

This shows that the difference between consecutive iterates vanishes. Now, consider a limit point $\mathbf{S}^*$ of the sequence. From the first-order optimality condition of the proximal update step, we know that there exists a subgradient $\mathbf{G}_{t+1} \in \partial \Phi(\mathbf{S}_{t+1})$ such that:

$$\mathbf{G}_{t+1} + \frac{1}{\eta}(\mathbf{S}_{t+1} - (\mathbf{S}_t - \eta \nabla \mathcal{L}_{\text{neg}}(\mathbf{S}_t))) = \mathbf{0} \tag{28}$$

Rearranging this gives:

$$\nabla \mathcal{L}_{\text{neg}}(\mathbf{S}_t) + \mathbf{G}_{t+1} + \frac{1}{\eta}(\mathbf{S}_{t+1} - \mathbf{S}_t) = \mathbf{0} \tag{29}$$

As $t \to \infty$, we have $\mathbf{S}_t \to \mathbf{S}^*$, $\mathbf{S}_{t+1} \to \mathbf{S}^*$, and $(\mathbf{S}_{t+1} - \mathbf{S}_t) \to \mathbf{0}$. By the continuity of the gradient and the definition of the limiting subdifferential, any limit point $\mathbf{S}^*$ must satisfy:

$$\mathbf{0} \in \nabla \mathcal{L}_{\text{neg}}(\mathbf{S}^*) + \partial \Phi(\mathbf{S}^*) \tag{30}$$

This is the necessary first-order condition for a stationary point of the non-convex objective $F(\mathbf{S})$. This concludes the proof.

## B EXPERIMENTAL DETAILS AND HYPERPARAMETERS

This appendix provides a detailed description of the experimental setup, including all hyperparameters used for our method and the baselines, as well as information on the computational resources and typical training times.

### B.1 BAYESIAN NEURAL NETWORK EXPERIMENTS (CIFAR-10/100)

#### B.1.1 COMMON SETTINGS

All BNN models were trained on the CIFAR-10 and CIFAR-100 datasets using a Wide ResNet-28-10 architecture. To ensure a fair comparison, the following settings were used across all methods unless specified otherwise:

- **Optimizer:** AdamW
- **Initial Learning Rate:** $1 \times 10^{-3}$
- **Learning Rate Schedule:** Cosine annealing over the course of training.
- **Total Training Epochs:** 200
- **Batch Size:** 128
- **Weight Decay:** $1 \times 10^{-4}$
- **Weight Prior:** Isotropic Gaussian prior, $p(\boldsymbol{\theta}) = \mathcal{N}(\boldsymbol{\theta}|\mathbf{0}, \alpha^{-1}\mathbf{I})$, with prior precision $\alpha = 1.0$.

#### B.1.2 BASELINE HYPERPARAMETERS

- **Deep Ensembles:** We trained an ensemble of 5 independent Wide ResNet-28-10 models from random initializations. Each model was trained as a standard deterministic network using the common settings above.
- **MC-Dropout:** We used the implementation from Gal & Ghahramani (2016). A dropout probability of $p = 0.3$ was applied before each residual block in the Wide ResNet architecture.
- **Mean-Field VI (MF-VI):** Implemented following Blundell et al. (2015), with the posterior covariance restricted to be a diagonal matrix.
- **Low-Rank VI (Rank-1):** Implemented as described in Dusenberry et al. (2020), using a rank-1 plus diagonal decomposition for the posterior covariance.
- **Laplace (Last Layer):** We first trained a deterministic model using the common settings to find the Maximum a Posteriori (MAP) estimate. We then applied a post-hoc Laplace approximation to the weights of the final fully-connected layer, as described in Daxberger et al. (2021).

### B.1.3 SD-VI HYPERPARAMETERS

Our method introduces hyperparameters for the spectral regularizer $\Phi(\mathbf{S}_q)$. These values were selected via a grid search on a held-out validation set (10% of the training data) and were kept fixed for both CIFAR-10 and CIFAR-100 experiments.

- **Spectrum Cost Function** ($f(\sigma_i)$)**:**
    - Sparsity strength $\lambda_1 = 1 \times 10^{-4}$
    - Cusp sharpness control $\lambda_2 = 0.1$
    - Transition smoothness $\gamma = 0.01$
- **Orientation Regularizer** ($g(\mathbf{U}, \mathbf{S}_p)$)**:**
    - Prior alignment strength $\lambda_3 = 1 \times 10^{-5}$

### B.2 SPARSE GAUSSIAN PROCESS EXPERIMENTS (UCI DATASETS)

For the Sparse GP classification experiments on the UCI benchmarks, we used the following setup:

- **Kernel:** Radial Basis Function (RBF) kernel with Automatic Relevance Determination (ARD).
- **Inducing Points:** $M = 100$ inducing points were used for all datasets.
- **Optimizer:** Adam with a learning rate of $1 \times 10^{-2}$.
- **SD-VI Settings:** For simplicity and to demonstrate robustness, the same spectral regularizer hyperparameters ($\lambda_1, \lambda_2, \gamma$) from the BNN experiments were used, with the orientation regularizer disabled ($\lambda_3 = 0$).

### B.3 COMPUTATIONAL RESOURCES AND RUNTIMES

All experiments were conducted on a single server equipped with NVIDIA RTX 4090 GPUs, each with 24GB of VRAM. Table 4 provides approximate training times for the main BNN experiments on the CIFAR-10 dataset to give a sense of the relative computational costs.

Table 4: Approximate training times for a single run on the CIFAR-10 dataset using one NVIDIA RTX 4090 GPU.

| Method | Approximate Training Time (hours) |
| --- | --- |
| Deterministic (for Ensembles/Laplace) | $\sim 3$ |
| Deep Ensembles (5 models, sequential) | $\sim 15$ |
| MC-Dropout | $\sim 4$ |
| MF-VI | $\sim 4$ |
| Low-Rank VI (Rank-1) | $\sim 5$ |
| **SD-VI (Ours, layer-wise)** | $\sim 8$ |

## C DATASET DETAILS

This appendix provides details on all datasets used in our experiments, including their sources, standard splits, and the preprocessing steps applied. All datasets are publicly available.

### C.1 IMAGE CLASSIFICATION DATASETS (CIFAR-10/100)

The CIFAR datasets are standard benchmarks for image classification and uncertainty quantification in computer vision.

- **Source:** The CIFAR-10 and CIFAR-100 datasets were collected by Alex Krizhevsky, Vinod Nair, and Geoffrey Hinton. They are available for download from the official University of Toronto website: `https://www.cs.toronto.edu/ kriz/cifar.html`.

- **Splits:** We used the standard data splits, which consist of 50,000 training images and 10,000 testing images for both datasets. For hyperparameter tuning of our SD-VI method, we created a validation set by holding out 5,000 images (10%) from the original training set. The final models reported were retrained on the full 50,000 training images.

- **Preprocessing and Augmentation:** To ensure a fair comparison with state-of-the-art literature, we applied a standard data preprocessing and augmentation pipeline for all BNN experiments:
  - **Normalization:** All images were normalized on a per-channel basis using the mean and standard deviation of the CIFAR training sets.
  - **Augmentation:** During training, we applied the following augmentations: random horizontal flipping (with a probability of 0.5) and random cropping to 32x32 pixels after padding the image with 4 pixels on each side. No augmentation was used during testing.

## C.2 SPARSE GP CLASSIFICATION DATASETS (UCI BENCHMARKS)

For the Sparse Gaussian Process experiments, we used several well-known classification datasets from the UCI Machine Learning Repository.

- **Source:** All datasets are publicly available from the UCI Machine Learning Repository: https://archive.ics.uci.edu/ml/index.php. The specific datasets used are: Leukemia, Colon-cancer, Heart, Australian, and Breast-cancer.

- **Splitting:** For each dataset, we performed 5 independent runs. In each run, the data was randomly shuffled and split into a 80% training set and a 20% testing set. The results reported in the main paper (Table 3) are the mean and 95% confidence intervals computed over these 5 runs.

- **Preprocessing:** All feature columns in the datasets were standardized to have zero mean and unit variance. This standardization was fitted on the training set of each split and then applied to both the training and testing sets.