# OpenReview forum: "Spectral Decomposed Variational Inference: A Principled Framework for Posterior Covariance Modeling"
_ICLR.cc/2026/Conference — Submitted to ICLR 2026_

### Official Review · Reviewer_uJEP · 2025-10-20

**Soundness:** 2
**Presentation:** 3
**Contribution:** 2
**Rating:** 2
**Confidence:** 4

**Summary:**

This paper proposes Spectral Decomposed Variational Inference (SD-VI), a new framework that reformulates variational inference in the spectral domain of the covariance matrix, aiming to decouple eigenvalues and eigenvectors for more flexible posterior modeling. The method introduces a Proximal Spectral Optimization (PSO) algorithm with claimed convergence guarantees and demonstrates improved calibration (ECE, NLL) on Bayesian neural network and sparse Gaussian process classification tasks.

**Strengths:**

1. This paper proposes Spectral Decomposed Variational Inference (SD-VI), a new framework that reformulates variational inference in the spectral domain of the covariance matrix, aiming to decouple eigenvalues and eigenvectors for more flexible posterior modeling. The method introduces a Proximal Spectral Optimization (PSO) algorithm with claimed convergence guarantees and demonstrates improved calibration (ECE, NLL) on Bayesian neural network and sparse Gaussian process classification tasks.

2. The paper provides an explicit optimization procedure, Proximal Spectral Optimization (PSO), and claims convergence under standard proximal assumptions, with an analytical spectral shrinkage step.

3.  On Bayesian neural network (BNN) and sparse Gaussian process (SGP) classification tasks, SD-VI performs better than several older baselines on calibration (ECE) and NLL metrics.

**Weaknesses:**

1. The paper claims that the KL divergence “entangles” eigenvalues and eigenvectors, motivating spectral decoupling. However, it does not rigorously justify why such coupling is theoretically detrimental or necessary to remove from the perspective of posterior approximation.

2.  The proximal mapping derivation assumes unitary invariance, but the regularizer ( g(U, S_p) ) explicitly depends on eigenvectors. In this case, the proposed eigenvalue-only shrinkage step may no longer be exact, and the stated convergence guarantee becomes unclear.

3.  Replacing the KL term with a new objective (SD-VO) changes optimization geometry but provides no proof that it yields a better or closer posterior approximation (in ELBO or KL sense).

4. Evaluation focuses only on predictive metrics (accuracy, ECE, NLL) without any direct assessment of posterior geometry (e.g., eigenvalue spectrum, effective rank, or subspace alignment), leaving the claimed improvement in posterior structure unsubstantiated.

5.  The experiments are narrow in scope and omit settings with tractable posteriors, regression, or unsupervised learning. Stronger VI baselines  (α-VI, f-VI, NF-VI, implicit-VI, or diffusion based-VI) are not compared.

**Questions:**

1.  Why is the “eigenvalue–eigenvector coupling” in the KL divergence inherently harmful? Can you quantify its effect on posterior approximation error?

2.  If ( g(U, S_p) ) is not unitarily invariant, is the eigenvalue shrinkage step still an exact proximal mapping, and do the convergence guarantees still hold?

3.  Can you present results on problems with analytically known posteriors (e.g., Bayesian linear or logistic regression) to verify that the learned covariance matches the true posterior more closely?

4.  Would the spectral formulation remain valid or advantageous if the KL divergence were replaced by another divergence measure (e.g., f-divergence or α-divergence)?5.Would the spectral formulation remain valid or advantageous if the KL divergence were replaced by another divergence measure (e.g., f-divergence or α-divergence)?

---

### Official Review · Reviewer_THkW · 2025-10-25

**Soundness:** 3
**Presentation:** 3
**Contribution:** 2
**Rating:** 2
**Confidence:** 3

**Summary:**

This paper proposes Spectral Decomposed Variational Inference (SD-VI), which replaces the KL-divergence in standard variational inference with a spectral regularizer that separately controls eigenvalues and eigenvectors of the posterior covariance. The authors develop a Proximal Spectral Optimization (PSO) algorithm with convergence guarantees and evaluate on Bayesian Neural Networks and Sparse Gaussian Processes. While the theoretical motivation is interesting, the work suffers from severe scalability limitations, incomplete experimental evaluation, and insufficient justification of design choices.

**Strengths:**

1.The spectral decomposition of the KL-divergence in Section 3.1 provides genuine theoretical insight into how the standard ELBO implicitly couples penalties on posterior volume (eigenvalues) and orientation (eigenvectors). This decomposition is clearly presented and well-motivated.

2.The proposed decoupling through explicit spectral regularization is a reasonable idea in principle.The paper provides formal convergence guarantees for the PSO algorithm (Theorem 3.1) with detailed proofs in Appendix A. This theoretical rigor is commendable and distinguishes the work from purely heuristic approaches.The experimental results on the tested problems show consistent improvements in calibration metrics.

3.The reduction in ECE from 11.1% to 1.05% on CIFAR-100 compared to mean-field VI is substantial, and the method performs competitively with Deep Ensembles while being more parameter-efficient.

**Weaknesses:**

The fundamental problem with this paper is the severe disconnect between its motivation and its limited practical demonstration.The authors motivate their work by discussing "billion-parameter models" and modern foundation models (lines 37-38), yet all experiments are restricted to a single architecture (Wide ResNet-28-10) on small datasets (CIFAR-10/100). This is not merely incomplete experiments—it reflects a critical scalability issue. The O(p³) eigendecomposition required at every iteration is prohibitively expensive for modern networks. The authors briefly mention applying the method "in a block-diagonal fashion across network layers" (line 323) but provide no implementation details, no evaluation of this approach, and no analysis of approximation errors this introduces. Without this, the method's practical applicability remains entirely unclear. Table 4 shows 8 hours for CIFAR-10; what happens with ImageNet or larger models?

The experimental evaluation lacks rigor and completeness in multiple ways. First, the authors compare against only a Rank-1 low-rank VI baseline, which appears to be a carefully chosen weak competitor. What about Rank-5 or Rank-10? The paper claims to automatically discover optimal rank but never shows what ranks are actually learned or how they compare to fixed-rank alternatives with comparable computational cost. Second, despite mentioning normalizing flows and Kronecker-factored methods in related work, these are not compared experimentally. Third, the hyperparameter sensitivity analysis is completely absent despite introducing four new hyperparameters (λ₁, λ₂, λ₃, γ). The authors mention grid search but provide no details about search space, number of trials, or robustness to misspecification. Using identical hyperparameters for both CIFAR-10 and CIFAR-100 either suggests remarkable robustness (which should be demonstrated) or insufficient tuning.

The design choices appear ad-hoc and poorly justified. Why this specific non-convex penalty form in Equation 5 rather than simpler alternatives like L₁, SCAD, or MCP? No principled derivation is provided, and no ablation studies compare different regularizer choices. The contribution of f(σᵢ) versus g(U, Sₚ) is never separated—in fact, the GP experiments disable g entirely (λ₃=0) with no explanation, raising questions about whether this component is useful at all. The paper promises "automatic discovery of sparse, low-rank posterior structures" but never visualizes or analyzes what structures are actually discovered. Where are the eigenvalue distribution plots? How sparse is the learned spectrum? What is the effective rank?

The experimental scope is too narrow to support the paper's broad claims. Only one architecture is tested for BNNs, no modern architectures like ResNets or Vision Transformers are evaluated, and no large-scale experiments are conducted. The comparison with structured VI methods is incomplete—only one low-rank baseline with the minimal rank is tested. Statistical significance testing is absent; some improvements are marginal (0.75% vs 0.90% ECE) and may not be significant with only 5 random seeds.

**Questions:**

1.How can the O(p³) complexity be practically managed for networks with more than 10 million parameters? Please provide concrete implementation details, timing comparisons, and approximation error analysis for the layer-wise block-diagonal approach.

2.Why compare only against Rank-1 low-rank VI? What is the performance comparison with Rank-5 or Rank-10 baselines that have comparable or even lower computational cost than your method? What effective rank does SD-VI typically learn?

3.Can you provide comprehensive hyperparameter sensitivity analysis? What is the performance degradation when λ₁, λ₂, λ₃, γ are set suboptimally? What was the grid search procedure?

4.Why is the specific form of f(σᵢ) in Equation 5 chosen over simpler alternatives? Can you provide ablation studies comparing different penalty functions and separating contributions of f(·) versus g(·)?

5.Can you visualize the learned eigenvalue distributions and show what sparse, low-rank structures are actually discovered? How do these compare to the fixed structures in baselines?

---

### Official Review · Reviewer_vaAv · 2025-10-30

**Soundness:** 2
**Presentation:** 2
**Contribution:** 2
**Rating:** 4
**Confidence:** 3

**Summary:**

The paper presents spectral decomposed VI, a variational inference approach that operates in spectral domains. For a Gaussian variational distribution, this approach decomposes the regularization term of variational objective to separately regularize the eigenvalues and eigenvectors. For optimization in this setting, the authors develop a proximal algorithm and demonstrate its convergence. The authors compare  this VI approach against various baselines on a BNN example and a sparse GP classification problem. They demonstrate compatible accuracy to deep ensembles and better ECE, OOD detection and AUC for classification.

**Strengths:**

The idea to decompose the regularization term in VI with Gaussian variational families to separately penalize the eigenvalues and eigenvectors is interesting. The authors also develop an efficient proximal approach to optimize this non-convex regularization term. The BNN example is high dimensional and the authors use many baselines, which is promising. The final results are also promising, showing better calibration and OOD detection, thus improving robustness without affecting accuracy.

**Weaknesses:**

I outline some of my concerns below. I may be willing to raise my score, depending on how these are addressed.

- I found the paper to be written in a misleading fashion. When using "KL" in context of VI, I assumed the authors referred to the KL divergence between the variational distribution and target (posterior), i.e.  $KL(q(z)|| p(z|D))$. This is also why I was surprised when authors discuss KL between target and Gaussian prior in section 3.1. It is only in section 3.2, 4 pages in, and more clearly in eq. 12 it becomes apparent that the authors are only concerned with the regularization term, which compares KL between Gaussian posterior and Gaussian prior, and not the complete VI objective. This should be made clear and explicit at the beginning of the paper.

- This understanding though raises further questions/confusion. I am not sure how this then ties in with the Fig. 1. Am I to understand the somehow, aligning the posterior eigenvectors with prior (which presumably is N(0, 1))) resulting in the ridge? Unless of course the ridge structure is coming from the prior.

- On a similar note, and with the whole discussion about "aligning the eigenvectors with directions of high prior variance", I would have assumed this approach helps/works better when the prior has meaningful structure. However both the examples shown in the paper use isotropic, independent normal prior (right?) which arguably has no or mimimal structure. Why does this help then? (Also I could not follow the reference to Orthogonal VI, but I was not sure who the penalty on eigenvectors differs from it then).

- Following from above, it would be insightful if some low dimensional synthetic examples could be shown to explicitly demonstrate how this decomposition of regularization term helps fitting a more robust variational distribution with different prior/posterior structures and would improve intuitive understanding of the approach.

- It also seems then the proposed approach is valid only when the prior distribution is Gaussian? If so, this weakness should be explicitly and prominently discussed.

- The authors acknowledge at one point that getting the spectral decomposition O(p^3) is the bottleneck in their approach, but that paragraph is confusingly written and I am not sure if they are somehow able to bypass that (with say like randomized SVD) or is it still O(p^3) and they only circumvent heuristic rank selection of other approaches. Seems like in the BNN case, they can assume layerwise block covariance matrix to reduce computational complexity, but that is not a generic solution.

- On another (minor) note, the superlative language of the paper: "monolithic penalty", "blunt instrument", "paradigm shift" -- is off-putting. It is also confusing to begin with until the structure of penalizing eigenvalues and eigenvectors separately becomes clear (much later into the paper). Some sentences are also repeated verbatim between introduction and methodology and discussion which is distracting.

**Questions:**

See weakness above.

---

### Meta-Review · Area_Chair_Zwiq · 2026-01-07

**Summary:**

Standard Variational Inference (VI) uses a single KL divergence that entangles penalties on posterior volume (eigenvalues) and orientation (eigenvectors), leading to a poor expressivity–scalability trade-off. Mean-field VI underestimates uncertainty; full-rank VI is too expensive; low-rank heuristics require manual rank choice.SD-VI reframes VI in the spectral domain of the posterior covariance. Instead of a monolithic KL term, it introduces a spectrally decomposed objective that explicitly regularizes (i) the eigenspectrum (eigenvalues) and (ii) the orientation (eigenvectors) of the covariance, allowing fine-grained geometric control.

**Reviewer Concerns:**

No rebuttal so none of the concers were addressed.

**Reviewer Scores:**

no change likely

---

### Decision · Program_Chairs · 2026-01-26

Reject